# Modeling nearshore-offshore water exchange in Lake Ontario

**Bogdan Hlevca**[1]*, **Edward Todd Howell**[1], **Reza Valipour**[2], **Mohammad Madani**[3]

**1** Ministry of the Environment, Conservation and Parks, Toronto, Canada, **2** Environment and Climate Change Canada, Burlington, Canada, **3** DHI Water and Environment Inc, Cambridge, Canada

* bogdan.hlevca@ontario.ca

**Data Availability Statement:** All relevant input data sources are listed within the paper and its Supporting Information file. The datasets from the ministry's field measurements used for calibrating the model in this study https://doi.org/10.5281/

## Abstract

The water quality and resources of Lake Ontario's nearshore ecosystem undergo heightened stress, particularly along the northwest shoreline. Hydrodynamic processes linking the distinct nearshore and offshore trophic structures play a crucial role in transporting nutrient-loaded water along and across the shore. Despite the pivotal connection between algae growth and the development of nuisance proportions, the scales over which these processes operate remain poorly understood. This study delves into the exchange dynamics between nearshore and offshore areas of Lake Ontario throughout 2018, employing a validated three-dimensional numerical model. A virtual passive age tracer is utilized to discern horizontal mixing time scales between nearshore regions of the lake (water depth < 30 m) and offshore locations. The dispersal pattern, as revealed by a passive tracer released from eight points around the model lake's perimeter, indicates more extensive diffusion in late summer when lake-wide stratification is established, compared to the mixed period. Coastal upwelling events, leading to intrusions of hypolimnetic waters, significantly contribute to net cross-shore transport, with the most pronounced effects observed in May and June when the offshore thermocline is shallow. In the northern part of the lake, dispersal predominantly occurs alongshore, mirroring the prevailing cyclonic (counterclockwise) coastal circulation during the stratified season. This pattern is a consequence of a 45% increase in upwelling events compared to three decades ago. In the northwestern and southern sectors of the lake, elevated cross-shore mixing is attributed to geomorphology-induced cross-basin currents.

## Introduction

The nearshore areas of large bodies of water contain the richest biota and are the most valuable water resources for the adjacent communities. Unfortunately, their proximity to dense urban communities, makes them also the most vulnerable to degradation [1]. In these lake areas, the horizontal mixing processes control the dispersion of factors (pollutants, nutrients, and sediment) that determine the water quality [2]. Therefore, understanding the processes and factors that affect the horizontal mixing and the scales over which these processes occur, is important for predicting the evolution of the biosystems and of future events.

zenodo.8306165 in the Zenodo Repository. The datasets generated by the model during the calibration and validation phases are openly available in open access at https://doi.org/10.5281/zenodo.8306884 in the Zenodo Repository. The datasets generated during the analysis phase are available from the corresponding author on reasonable request.

**Funding:** The author(s) received no specific funding for this work.

**Competing interests:** The authors have declared that no competing interests exist.

Lake Ontario is the most eastward of the Laurentian Great Lakes, the smallest by surface area (19,010 km$^2$; ranks 14th largest in the world), and the second smallest by volume (1,640 km$^3$), after Lake Erie. Its length is 311 km, the average and maximum depths are 86 and 244 m, respectively [3]. The average hydraulic retention time is between 6.4 and 9 years [4], which is shorter than the other Laurentian Great Lakes (except for Lake Erie), but long enough to determine a lag in response to changes in management such as nutrient control strategies [5]. The main discharge into the lake is from Lake Erie, through the Niagara River [6, 7] and constitutes the headwater for the St. Lawrence River.

Lake Ontario's native ecosystem supports several native species, such as walleye (*Sander vitreus)*, Coho salmon *(Oncorhynchus kisutch)*, Chinook salmon *(Oncorhynchus tshawytsch)* and several trout varieties, including rainbow (*Oncorhynchus mykiss*) and steelhead (*Oncorhynchus mykiss irideus*) [8]. Invasive mussels have succeeded to cover much of the bottom of the lake in the coastal areas. They are benthic filter feeders capable of attaching to hard substrates and due to the magnitude of their biomass and filtering activity they changed the nutrient cycle in the lake, which has significant implications in determining the water quality, especially in the nearshore [9]. Nonetheless, the benthic cover in the offshore is significant and increasing [10].

Lake Ontario's native ecosystem, due to its downstream location, is relatively impaired when compared to the upper Great Lakes [11]. Studies of the lake's food web using stable isotope analysis [12], and distributions of fish [13], benthic invertebrates [14, 15], and zooplankton [16] suggest that its nearshore and offshore components are roughly divided by the 30 m depth contour. The nearshore area is located within the coastal boundary layer (CBL) [17–19] and exhibits notable differences in hydrodynamics, water quality and biota when compared with the offshore. In addition, in recent decades, warmer summer surface temperature on the Laurentian Great Lakes and stronger meteorological events [20] have been linked to a reduction of ice cover trend [21]. As a result, the ranges for warm water species have extended toward north and are expected to further expand [22].

Previous studies (e.g., [23, 24]) have revealed seasonal variations in Lake Ontario's circulation due to its dimictic cycle involving vertical mixing and density stratification [5]. It is stronger in winter, predominantly anticyclonic (clockwise), akin to Lakes Superior, Michigan, and Huron [23], except for the Rochester Basin, where circulation is mainly cyclonic. In early spring and fall, cross-shelf transport is restricted during the onset and dissipation of the thermocline by the thermal bar [24]. After the stratification is established across the lake, the new formed cyclonic coastal circulation separates nearshore waters from the offshore [23, 24]. Consequently, this separation aids in concentrating nutrients and pollutants in the nearshore, exacerbating the nearshore shunt [9], and thereby promoting *Cladophora* algae growth. Moreover, we hypothesize that coastal upwelling and downwelling events [25, 26], along with instabilities associated with their thermal front, influence horizontal mixing, reducing the entrapment of the predominant coastal flow and enhancing the exchange between the nearshore and offshore areas of the lake. This, in turn, could potentially increase nutrient uptake for seasonal *Cladophora* growth [26, 27]. These perturbations of predominant processes are expected to affect the ecosystems in the nearshore by controlling the distribution of plankton, especially the phytoplankton [28] that is the main food source for *Dreissena* mussels (Rasmussen, personal communication, 2021). The influence of these perturbations over the nearshore ecosystems is not well understood.

The objective of this study was to investigate the exchange between the nearshore and offshore areas of Lake Ontario using a validated three-dimensional model, focusing on understanding the hydrodynamic processes that contribute to nutrient transport and its link to algae growth. We assessed the variability of horizontal exchanges in Lake Ontario using virtual

tracers, which allowed us to estimate the magnitude of the horizontal mixing in different areas of the lake. Furthermore, we evaluated the correlation between water current speeds, horizontal mixing, wind patterns, and water temperature during the stratified season, as well as the contributions of coastal upwelling events and geomorphology-induced cross-basin currents to the net cross-shore transport in the northern nearshore of Lake Ontario. Ultimately, the objective was to enhance our understanding of the processes influencing the exchange and dispersal of water, nutrients, and algae in Lake Ontario's nearshore and offshore ecosystems. The numerical model setup, and various model runs conducted in this study are briefly described in the following section. The results and discussion sections present the findings, and conclusions are drawn at the end.

## Methods

### Numerical model description

The currents and thermal structure of Lake Ontario were simulated using DHI's MIKE 3 modeling framework [28]. MIKE 3's hydrodynamic module is based on the numerical solution of the three-dimensional incompressible Reynolds-averaged Navier-Stokes equations, incorporating the assumptions of Boussinesq and hydrostatic pressure [29], alongside the conservation of mass and momentum in three dimensions of a Newtonian fluid. The resolution of turbulent fluctuations and the closure problem is achieved through the Boussinesq eddy viscosity concept, linking Reynolds stresses to the mean velocity field. This involves a combination of the Smagorinsky formulation for the horizontal direction and a k-ε formulation for the vertical direction. The model utilized a flexible unstructured mesh with over 15,000 triangular elements in the horizontal plane, consisting of 16 sigma layers up to a depth of 40 meters. Additionally, it included a set of 14 z-level layers for deeper areas in the vertical direction. The setup, calibration, and validation of the model are comprehensively detailed in Hlevca et al. [30] and are also partially described in the supplementary electronic material (S1-S8 Figs in S1 File).

### Modelling scenarios and definition of tracers

We assessed nearshore-offshore water exchanges and horizontal mixing timescales in the nearshore using two sets of modelled tracers throughout the entire calendar year 2018. The nearshore zone is defined as the coastal area with depths less than 30 m and is consistent with previous research (e.g., [31]). The first virtual tracer is the *water age* tracer (hereafter *age*), and it was used to estimate the residence time of the water in the nearshore. This tracer, which measures time spent by the water parcels in the nearshore area, was incremented for each time step within the nearshore area. For the offshore areas and tributaries, the *age* was kept at a constant zero [32]. *Age* values were incremented by a value equal to the time step in the nearshore, where no mixing due to exchange occurred, and calculated based on the exchange between adjacent grid cells according to the modelled water fluxes determined by the MIKE 3 model, where mixing occurred. *Age* could be reduced below or increased above the initial value if mixing occurred during the time step. For instance, mixing between two nearshore cells with different ages increased the age in the cell where the age was initially lower by more than the occurring time step. *Age* needed a constant setting of a zero-age value in the offshore. Since the MIKE3 hydrodynamic module lacks the capability to calculate age and differentiate between the nearshore and offshore zones, we used MIKE ECO Lab [29]. ECO Lab, a generic programmable ordinary differential equation solver, is typically used for water quality simulations, but it can also be used for general programming, including zone classification. Each mesh element

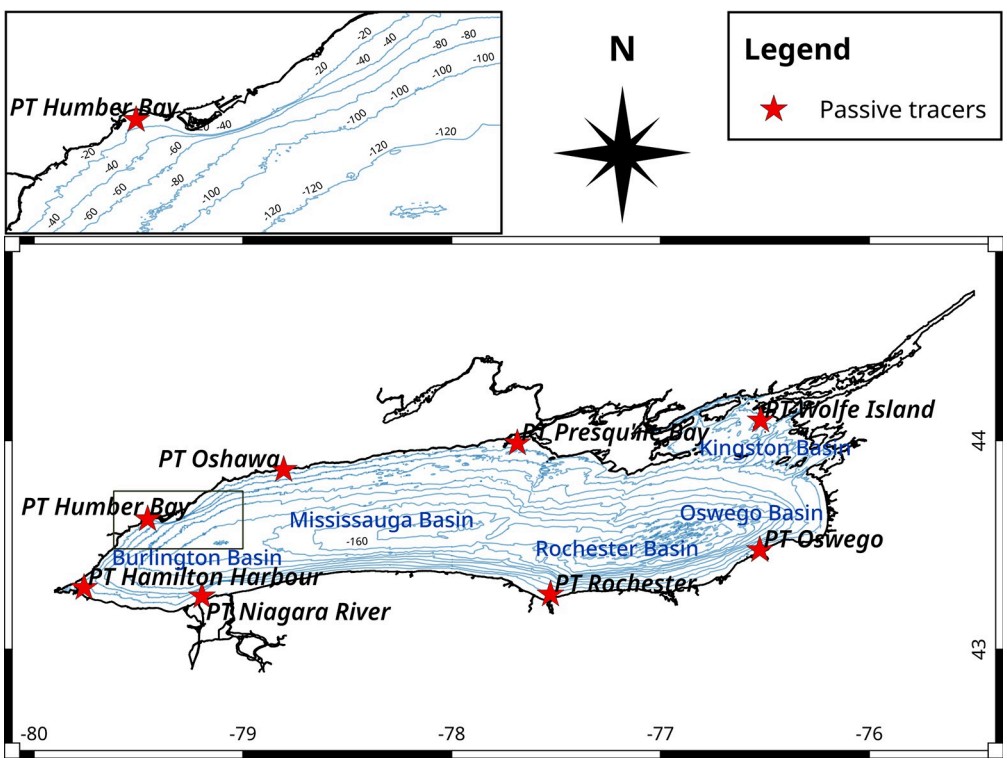

**Fig 1. Nearshore passive tracers' locations.** The nearshore points of release are where water depth is less than 30 m. Labelled points in the nearshore areas indicate the starting locations for passive dye tracers used in the second set of model runs. The depth at these locations varied between 10 m to 15 m.

in ECO Lab was assigned a value of 1 or 0 to distinguish between nearshore and offshore areas, respectively.

The second virtual tracer was designed as a nearshore regional tracer and was used to estimate the horizontal dispersion rates. We used two sets of the nearshore regional tracer (hereafter *NRT*), released at eight nearshore locations around the lake (Fig 1), following the approach by McKinney et al. [2]. The tracer concentration was set at 1x $10^6$ ppm and released vertically in a combined discharge of 2 $m^3$ $s^{-1}$ across all sigma layers, corresponding to one element of the horizontal mesh. The tracer discharge was well below the accuracy of water balance estimates for Lake Ontario, rendering an adjustment in the water balance unnecessary. The first *NRT* set was continuously released for 90 days, starting on April 1, and an identical second set released on July 1, also for 90 days. These *NRT* sets were used to assess alongshore and cross-shore dispersal under two lake conditions: 1) before and during the formation of stratification, and 2) when the entire lake was stratified. Cross-shore dispersal was defined as the change in the concentration of *NRT* in offshore regions, and alongshore dispersal was defined as the change in concentrations of the same tracer in nearshore regions [2]. Both *age* and *NRT* were used in the same first simulation.

The second simulation study (Table 1) used a more broadly based passive *nearshore tracer* (hereafter *NST*) with a constant concentration set in the nearshore at 100% and an initial concentration of 0% set in the offshore. *NST* was used to estimate the nearshore-offshore exchange. During the simulation, we expected that the concentration of *NST* in the offshore would increase due to mixing with the nearshore water. The MIKE 3 framework is able to keep constant conditions at the domain boundaries, but it is not natively equipped to keep

**Table 1. Virtual tracers and simulations.**

| Tracer | Simulation number | Simulation Type | Tracing Period | Purpose |
|---|---|---|---|---|
| *Age* | *1* | Eulerian transport | Year 2018 | Determine water age in the nearshore |
| *NRT* | *1* | Eulerian transport | 90 days: April 1, July 1 | Determine the cross and along-shore dispersion |
| *NST* | *2* | Eulerian transport | Year 2018 | Determine the nearshore-offshore exchange |
| *NST* | *3* | Eulerian transport | March–December 2018 | Determine the nearshore-offshore exchange |
| *NST* | *4* | Eulerian transport | June–December 2018 | Determine the nearshore-offshore exchange |
| *NST* | *5* | Eulerian transport | September–December 2018 | Determine the nearshore-offshore exchange |

constant values inside the computational domain. We circumvented this limitation by developing a second model with a mesh that contained only the offshore area of Lake Ontario (Fig 2), for which the surrounding open boundary conditions were provided from the output of the entire lake model the simulation (S1 Fig in S1 File). The offshore model was run in identical conditions as the lake-wide model. The open boundary condition for *NST* were set at a constant value of 100% and an initial value of 0% inside the offshore model domain. The offshore model allowed us to estimate the exchange of nearshore water with the offshore across the vertical transect positioned at 30 m depth contour.

Four simulations (2, 3, 4 and 5) were performed, each started in January with current speed set to zero and a uniform temperature of 4°C on the entire model domain, while the 100% *NST* concentration at the open boundary started at a different date of the year: January, March, June, and September, respectively. This allowed us to assess the horizontal mixing under different seasonal conditions. Meteorological data from the National Center for Atmospheric Research (https://rda.ucar.edu/datasets/ds094.1) was applied as model forcings for the entire simulation period. The average daily *age*, *NST* and lake state variables including temperature and current speed were simulated for the entire 2018 year and correlation was performed

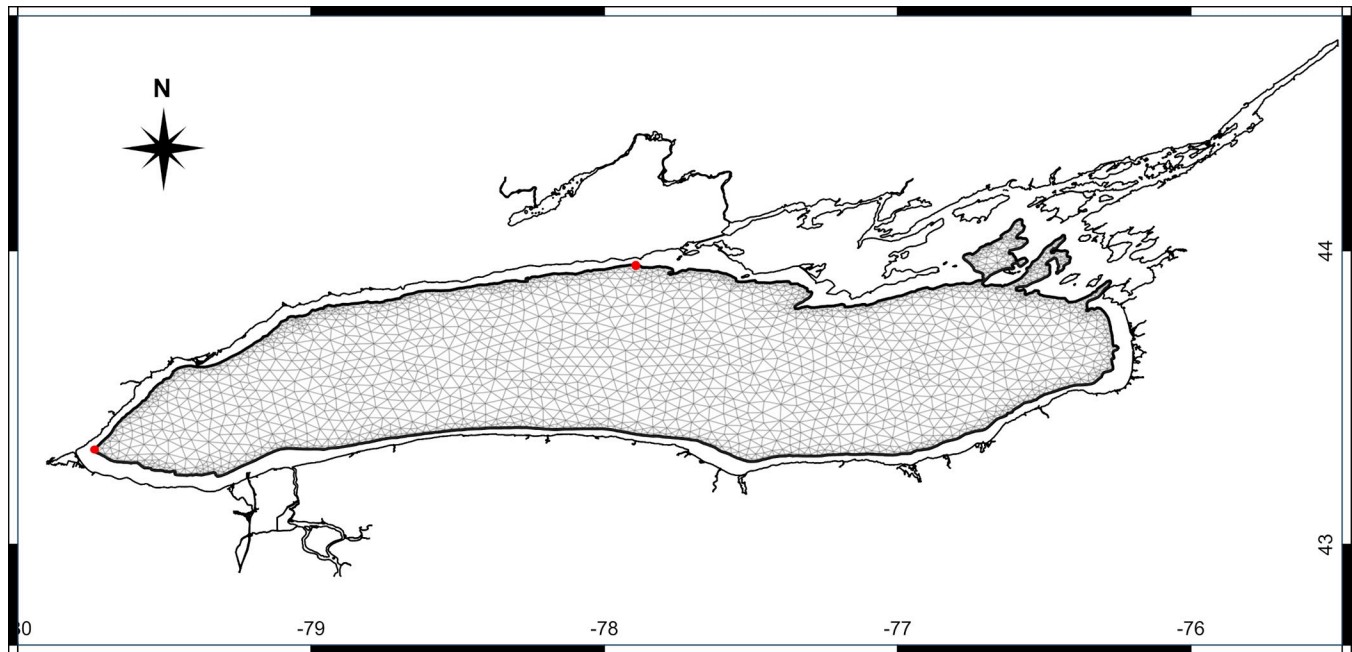

**Fig 2. Offshore modelling domain.** The thick black line represents the vertical transect at the 30 m isobath (Also see S1 Fig in S1 File). The red dots mark the beginning and the end of the vertical transect used to calculate the nearshore-offshore water exchange (from W to E clockwise).

for several inputs such as wind stress ($\tau_{wind} = \rho_{air} C_D U_h^2$) and air temperature. The simulations did not include ice coverage as 2018 was an ice-free year for Lake Ontario, with a short-lived exception in the Kingston Basin where partial ice-coverage lasted for a few days in mid-January.

## Results and discussion

### Model results

The model employed in this study underwent extensive calibration and validation, as detailed in Hlevca et al. [30]. It successfully resolved dominant physical processes, including currents, upwelling events (Fig 3 and S7 Fig in S1 File), near-inertial oscillations (S2-S7 Figs in S1 File), and internal waves. As highlighted in Hlevca et al. [30], we determined the vertical position of the upwelling front by fitting a line to the modelled station locations of the mean depth of the metalimnion layer. This method, thoroughly described by Austin and Barth [33] and applied by Jabbari et al. [26], was instrumental in our analysis. The dynamic nature of the temperature in the water column is graphically described in Fig 3 and the events listed in Table 2. More detail can be found the Supplementary electronic material and extensively documented in Hlevca et al. [30]

### Simulation of age

*Age*, spatially averaged between the five northern shore stations in the nearshore area (Fig 1), has generally steadily increased during the isothermal season, from January through March (~DOY 75), with similar patterns in top (blue lines, Fig 4A, left axis) and in bottom (orange lines, Fig 4A, left axis) layers. Periods of higher wind speeds favored sharp decreases in water *age* throughout the nearshore water column (e.g., DOY 80, 100 [30]). During the formation of the thermal bar (DOY 100–135) *age* remained virtually unchanged, oscillating around the same value (~2 days). In contrast, during the stratified season, even with lower wind speeds, the *age* patterns between the top layers and bottom layers were different. After DOY 135, the *age* in the surface layer, on average, fluctuated around the DOY 135 value until ~DOY 250, with significant downward variations during downwelling events, while the bottom layer *age* continued with high variability but with and increasing trend until the end of August (DOY 235), when stronger winds and the onset of the fall turn over favored increased both horizontal and vertical mixing. Between May to August (DOY 120–210), surface current speeds remained high despite of a slight decrease in wind speed (Fig 4B; S12 Fig in S1 File; and see Hlevca et al. [30]). During this period of stratification onset, a more stable water column was created (see Brunt-Väisälä values S8 and S9 Figs in S1 File). When the surface and bottom layers decouple, the wind energy is applied mostly in the surface layer, leading to a greater dispersal than in the bottom layer (Fig 4A), which is reflected in the initial higher increase of *age* in the bottom layer. During the mid fall period (DOY > 270), both surface and bottom layer *age* values decreased abruptly, most likely due the dissipation of the fall thermal bar, and due to an increase in the wind speeds. The seasonal increase and decrease of *age* are closely linked to the dimictic cycle of vertical mixing and water column stratification as observed in the modelled temperatures (dash lines, Fig 4A, right axis). During the period of *age* increase, or stagnation of the increase (shaded period in Fig 4A and 4B, DOY 137–250), the average water current speeds in the nearshore were on average at their lowest values of the entire year, while during the same period, the winds had few direction changes (S12 Fig in S1 File), the average direction was southwesterly (~ 220˚ azimuth). This may indicate that moderate winds with steady wind directions for longer periods of time have a greater influence on the overall water exchange

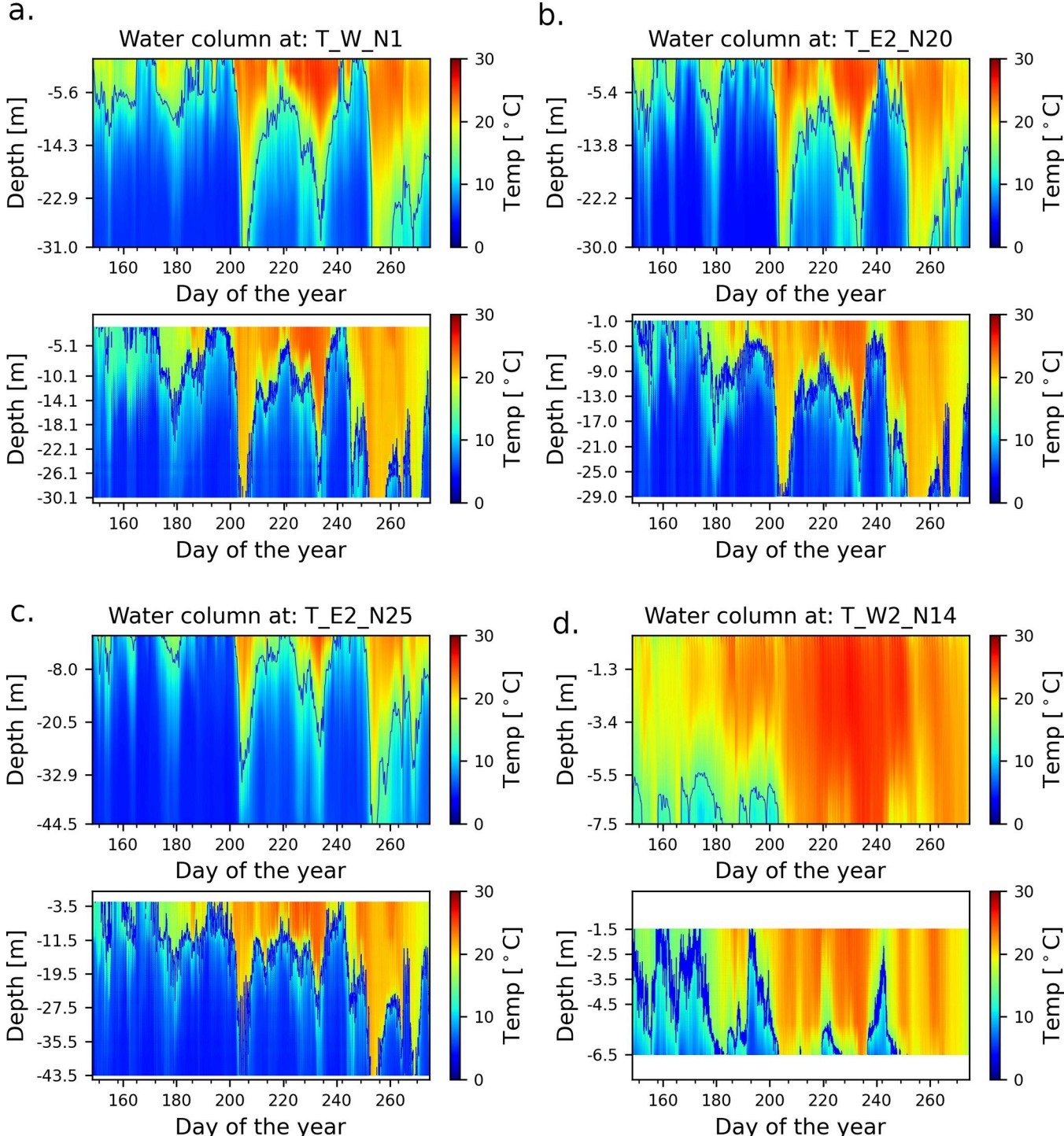

**Fig 3. Modelled and observed temperatures in 2018.** The upper plot in each sub-figure is the model result and the bottom plot is the observed data. Data from the western station W2_N14 is incomplete due to gaps in the observations in the top layers. a. Station W_N1; b. Station E2_N20; c. Station E2_N25; d. Station W2_N14. The figures have different colour bar scales to optimize for colour representation. Locations of the stations are shown in S11 Fig in S1 File.

between offshore and nearshore than stronger but short-lived wind bursts. In addition, upwelling and downwelling events were both correlated with significant drops of the age value (Fig 4A; see arrows; Table 2). Overall, the rate of *age* increase was lower than the slope of the

**Table 2. Details of observed upwelling and downwelling events between DOY 160 and 280 at station T_W_N1 (S11 Fig in S1 File.** Different locations have different event arrival times).

| Event | | | Wind | | | |
|---|---|---|---|---|---|---|
| No | Type | DOY | Duration [days] | Max speed [m/s] | Mean Speed [m/s] | Mean Direction 5 days average [˚] |
| 1 | upwelling | 170 | 3 | 6.8 | 4.3 | 211 |
| 2 | downwelling | 178 | 2 | 5.1 | 3.7 | 124 |
| 3 | upwelling | 184 | 3 | 5.7 | 5.1 | 262 |
| 4 | upwelling | 195 | 4 | 4.5 | 3.8 | 250 |
| 5 | downwelling | 204 | 4 | 9.4 | 6.1 | 70 |
| 6 | upwelling | 213 | 3 | 6.3 | 4.1 | 275 |
| 7 | upwelling | 220 | 2 | 4.7 | 3.8 | 248 |
| 8 | upwelling | 229 | 2 | 4.6 | 3.6 | 205 |
| 9 | downwelling | 233 | 2 | 6.2 | 4.9 | 78 |
| 10 | upwelling | 240 | 3 | 5.4 | 3.2 | 215 |
| 11 | upwelling | 244 | 2 | 5.2 | 3.1 | 202 |
| 12 | upwelling | 252 | 4 | 12.3 | 7.9 | 269 |
| 13 | downwelling | 270 | 2 | 10.2 | 7.1 | 109 |
| 14 | upwelling | 274 | 2 | 6.4 | 6.1 | 231 |

hypothetical 'no-exchange' aging line (Fig 4A, straight black dash-dotted line) indicating that the exchange between near shore and offshore areas occurred year-round.

## Simulation of nearshore regional tracers

The second part of the first simulation study consisted of using two identical sets of regional passive tracers (*NRT*) on the entire Lake Ontario mesh with 2018 data. The tracers with the greatest offshore dispersal were those on the western and northern end of the lake (Hamilton-Burlington Basin, Niagara, Humber Bay, and Presqu'ile Bay). The northern nearshore had a mostly along the shore dispersal while the western end of the lake cross-shore dispersal was present. In the southern nearshore and the eastern end of the lake the dispersal was lower but omnidirectional, except for the Oswego region, where the dispersal was high. Several factors may have contributed to this outcome, but arguably, the most important are the circulation patterns of the surface currents. Our simulations show a predominantly cyclonic spin (Fig 5A) in the upper Mississauga Basin during the stratified season, which although slightly deviates from observations made by earlier studies [5, 23, 34], it can be explained by a 45% increase in coastal upwelling events over the last thirty years [26]. These upwelling events can generate coastal jets induced by coastally trapped Kelvin waves [35], which have a cyclonic spin. Similar results were obtained by McKinney et al. [2] in Lake Superior, where the dispersion of the tracers released in the pre-stratified period was largely limited to the areas adjacent to their release points (Fig 5A), especially in the eastern part of the lake.

In the simulation that began on July 1, a notable increase in both, the alongshore, and the cross-shore dispersal was observed in the upper layers, particularly in the western end of the lake (Fig 5C). Stronger stratification during this period channeled the wind energy and density driven currents, on both the north and south shores, into a stronger surface cross-basin circulation (e.g., on the Toronto Harbour—Niagara cross-lake line). In contrast, due to the shielding of the thermocline in the bottom layers, the dispersal of the *NRT* was less extensive (Fig 5D) than in the simulation starting in April (Fig 5B). On the northern shore, the dispersal was predominantly alongshore from Presqu'ile Harbour to Hamilton, and the tracers released in the western region were well mixed in the entire western basin during the stratified season,

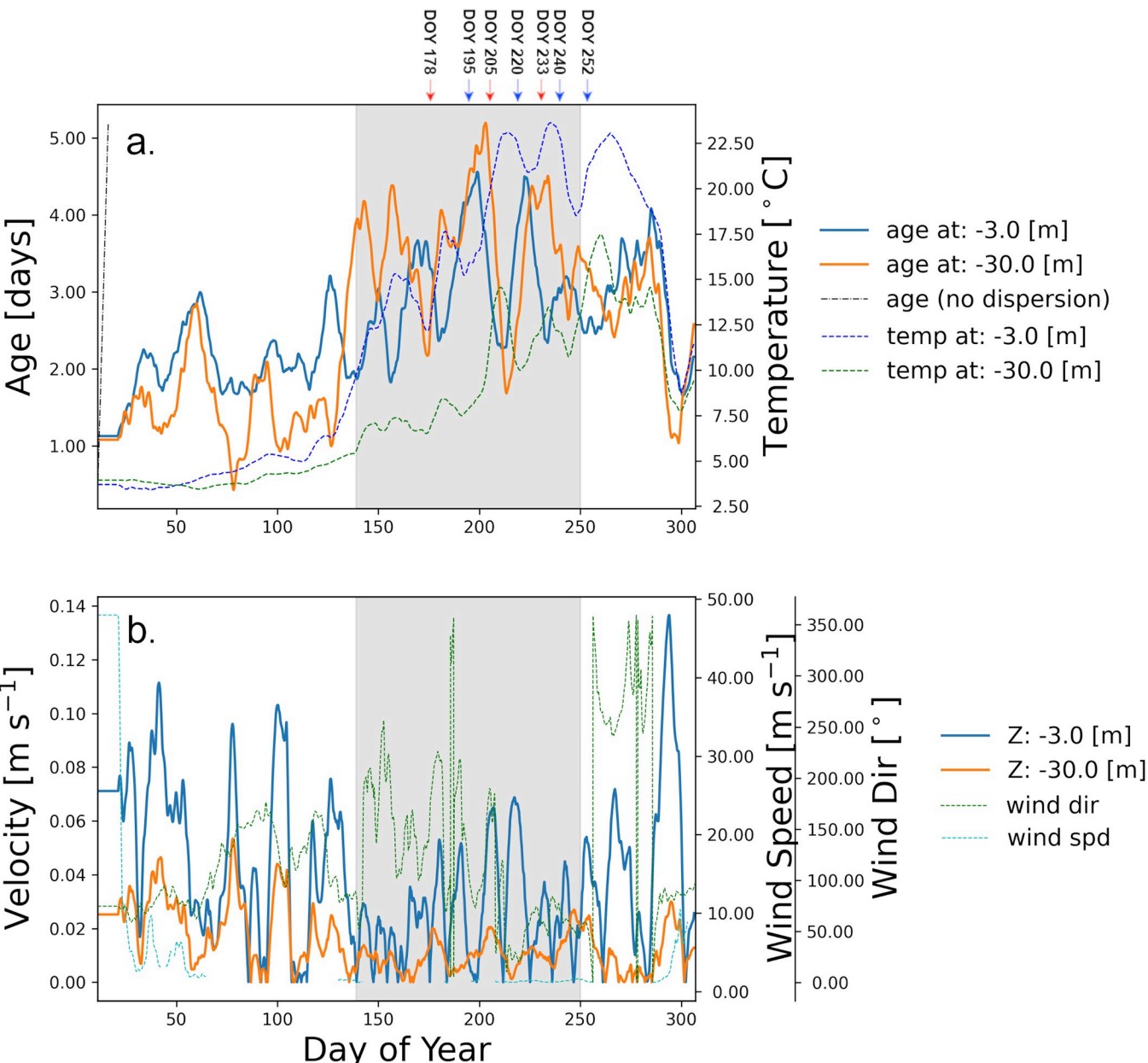

**Fig 4. Averaged age tracer, temperature, and velocity for the nearshore area.** (a) 10-day moving average *age* (blue for 3 m depth and orange solid lines for 30 m depth, left axis) and temperature (blue dashed lines for 3 m depth and green dashed line for 30 m, right axis). The black dash-dotted line is representing the 'no-exchange' aging line. Blue and red arrows mark selected upwelling and downwelling events, respectively. (b) The 10-day moving average velocity in the surface (blue) and bottom layers (orange). Wind speed (cyan dashed line) and direction (green dashed line) are superimposed. The values were calculated as 10-day moving average for the entire 2018 simulation averaging values for 5 northern shore locations of the 8 locations used to release the regional passive tracers: Hamilton, Humber Bay, Oshawa, Presqu'ile Bay, and Wolfe Island, (See Fig 1). Blue represents the E-W velocity and orange the S-N velocity. Dashed cyan and green are the wind velocity and direction. Shading indicates the time interval when *age* is not increasing.

when cross shore currents were more frequent. Tracers released in the eastern nearshore regions had a lower and omnidirectional dispersal.

The investigation into the dispersal of NRT suggests that the western end of the lake experiences higher dispersion rates compared to the northern shore, southern shore, and the eastern end, except, perhaps, for the Oswego zone. For instance, in the nearshore area east of Presqu'ile

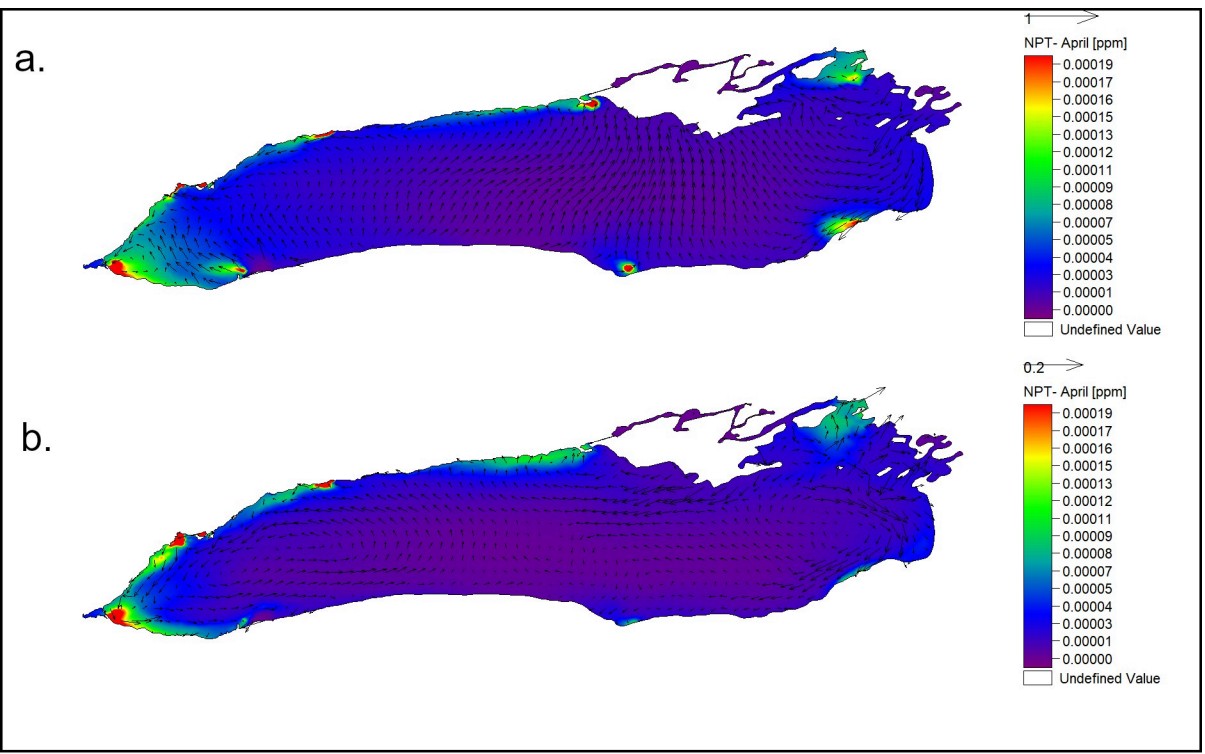

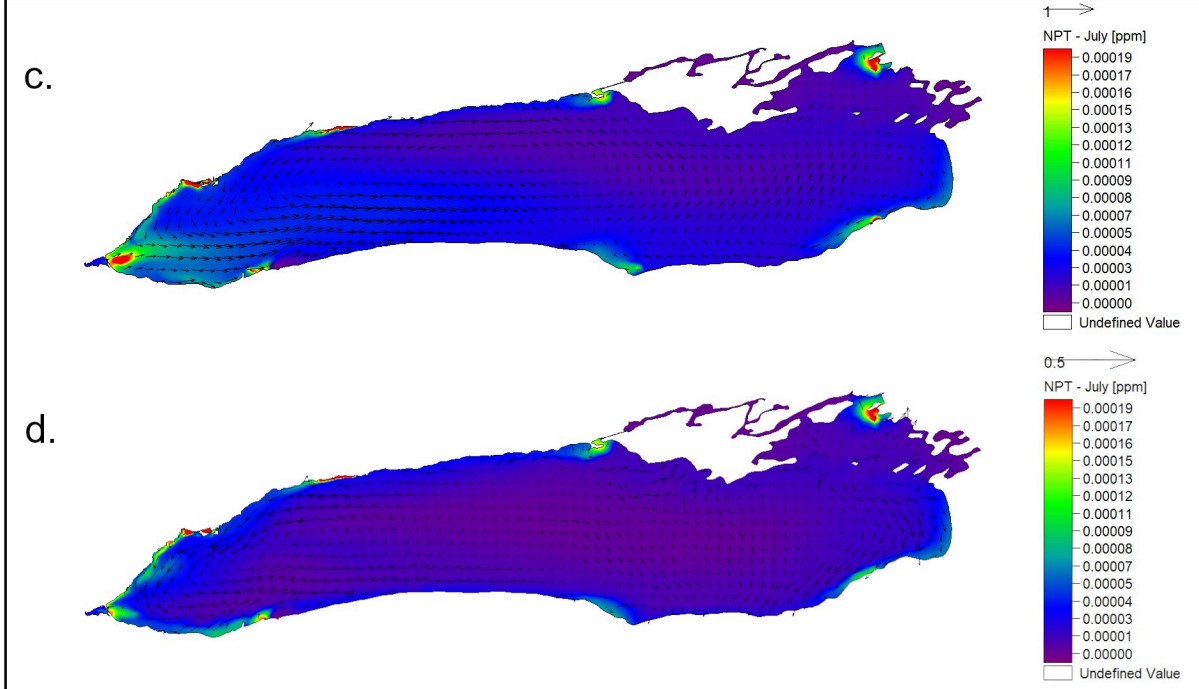

**Fig 5. Dispersal of *NRT*.** Passive tracer concentration after 62 days in the: (a) Spring-Early summer season top layer, passive tracers released on April 1. (b) Spring-Early summer season bottom layer, passive tracers released on April 1. (c) Late summer season top layer, passive tracers released on July 1. (d) Late summer season bottom layer, passive tracers released on July 1. Locations where tracers 1–8 were released are indicated in Fig 1. Vectors show snapshots of the direction of the currents at the end of the respective time intervals (62 days).

Bay, the mixing between nearshore and offshore is notably limited, making the nearshore water prone to aging. This finding aligns with the outcomes of earlier studies that delved into the circulation patterns in the eastern Lake Ontario [36, 37]. Cross-shore transport exhibited a significant increase during the second half of the year, likely attributable to rapid fluctuations and density-driven currents in the surface layer, influenced by wind energy primarily applied to the epilimnion. This resulted in lower concentrations of NST in the nearshore but higher concentrations in the upper layer offshore (Fig 5C).

### Simulation of nearshore passive tracers

The surface and bottom concentrations of the *NST* passive tracer in the offshore area were determined using the offshore sub-model (Fig 2 and S1 Fig in S1 File). We conducted four simulations, each starting at a different date with a 100% *NST* concentration boundary condition: January, March, June, and September (Fig 6). The results indicate that horizontal mixing is consistently more active in the surface layer. Specifically, we observed a steady and rapid increase in the *NST* concentration in the offshore region for both upper and bottom layers from January to April (Fig 6A and 6B). The nearshore-offshore exchange showed a further increase in the bottom layer during the month of September, while the upper layer appeared to be completely mixed (Fig 6A and 6B). Between the end of April and the beginning of May (DOY 106–120), there is a significant local minimum in the upper layer *NST* concentration plot and a corresponding upward peak in the bottom layer. This suggests a brief period of increased vertical mixing, possibly indicating the formation of the thermal bar.

Another notable observation in the plots is seen in the simulation of *NST* starting in September (Fig 6H). At DOY 261 (September 18), a reversed spike in the *NST* concentration is visible at a depth that matches that of the thermocline. This suggests that horizontal mixing in the thermocline is much slower than in the rest of the water column.

For the simulation with the tracer released in June (DOY 152) the surface concentration of *NST* continued to increase at a high rate, while the bottom concentration increased only very slowly until the end of August (Fig 6C). Similar to age, the difference between the dispersal of *NST* in the surface layer and the dispersal in the bottom layer during this interval (Fig 6C), can be attributed to the effect of stratification, which shields the bottom layers from wind action and prevents mixing in the hypolimnion. Time snapshots of the vertical *NST* concentration profiles (Fig 6E–6H) show that horizontal mixing is more pronounced in the upper part of the water column year-around. However, it is during the stratified period that the disparities in dispersal become more prominent (Fig 6G). The increases in *NST* were consistent with the changes in *age* (Fig 4A). At the end of the stratification period the concentrations of NST in the surface and bottom layers become closer again due to enhanced vertical mixing.

At the end of the year, the concentration of *NST* reached 100% in the top layer and close to that value in the bottom layer (Fig 6A), indicating that mixing between nearshore and offshore areas was completed within the span of a single year and even faster in the surface layers. We could not find modeling work in literature that estimated the timescale of nearshore-offshore exchange in Lake Ontario, and we hope that future work will be able to confirm our findings.

The *age* values discussed above (Fig 4) are lake wide averages, but the evolution of *age* at individual locations shows a variety of patterns (Fig 7). In the following discussion we analyze the values of *age* that were modelled at the 6 of the 8 nearshore locations where regional tracers have been released and that we considered representative for Lake Ontario (Fig 1). These locations exhibit shelfs of various widths and slopes and our goal was to determine the influence of shelf width and slope on the *age* values. In areas where the bottom slope (S10 Fig in S1 File) is moderate and the shelf is wider (Table 3; Hamilton—Burlington Basin, Wolfe Island—

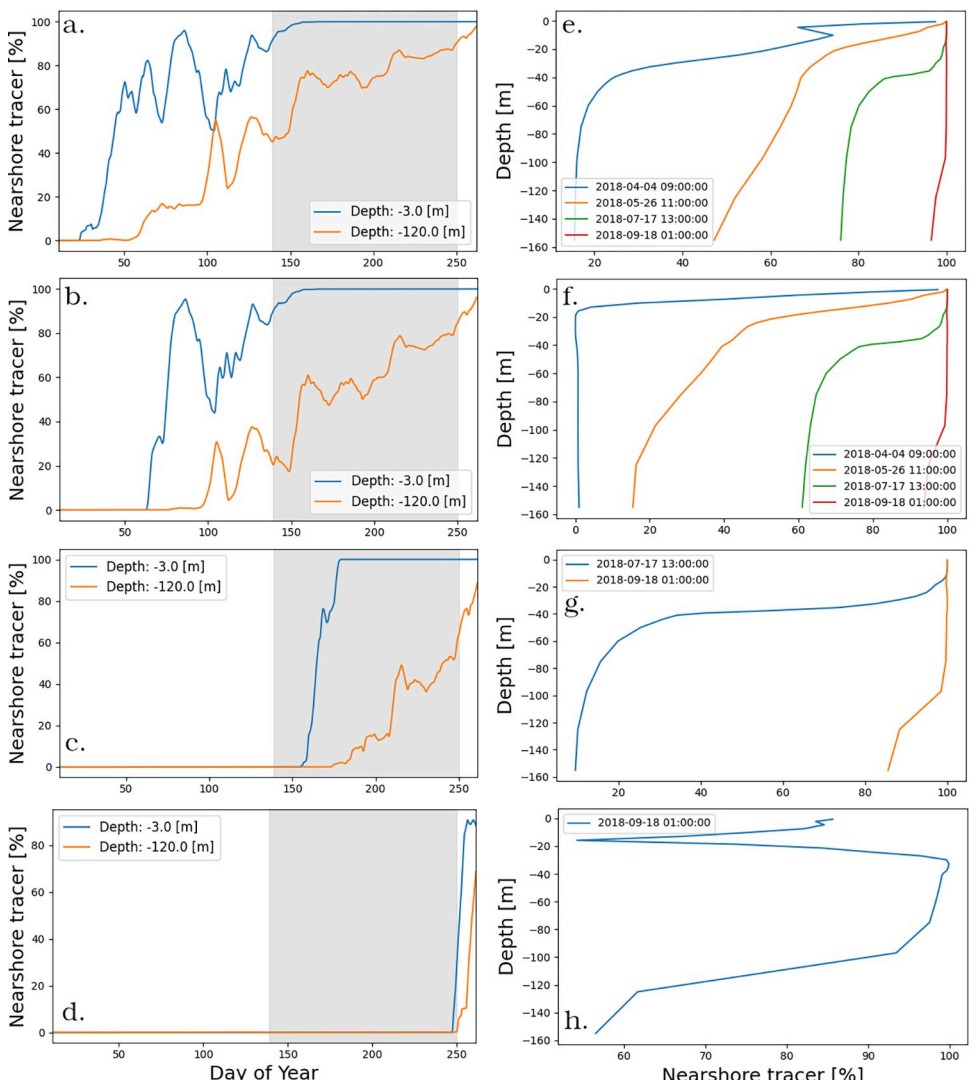

**Fig 6. The average *NST* tracer concentration in the offshore area (averaged over of eight points in the offshore of the lake) for four scenarios of the offshore sub-model.** (a) The *NST* tracer in the nearshore area has a constant 100% *NST* concentration starting from the beginning of the year (b) starting on March 1 (c) June 1, and (d) September 1. Figures (e), (f,) (g), and (h) show the vertical *NST* tracer concentration profiles snapshots corresponding to the simulation scenario on their left. Shading indicates the time when the average *age* is not increasing.

Kingston Basin, Presqu'ile Bay) *age* varied with the seasonal cycle of stratification, and the maximum annual *age* values reached over 10 days (Fig 7B, 7D, and 7E). In contrast, in areas where the shelf is narrow and steep (Table 3; Humber Bay, Rochester Basin, Oswego Basin (not shown)) maximum annual *age* was under 5 days (in Rochester nearshore even shorter ~ 1 day), indicating a stronger exchange between nearshore and offshore year around.

Determining horizontal mixing or dispersion directly requires time and instruments. Consequently, studies have looked at proxies to make predictions. McKinney et al. [2] have shown for Lake Superior that there is a good correlation between maximum *age* and average wind speed during the winter, which could be used as a good predictor for mixing between nearshore and offshore areas in the first part of the year. We took a slightly different approach for interpreting the mixing from the Lake Ontario model results for individual nearshore regions.

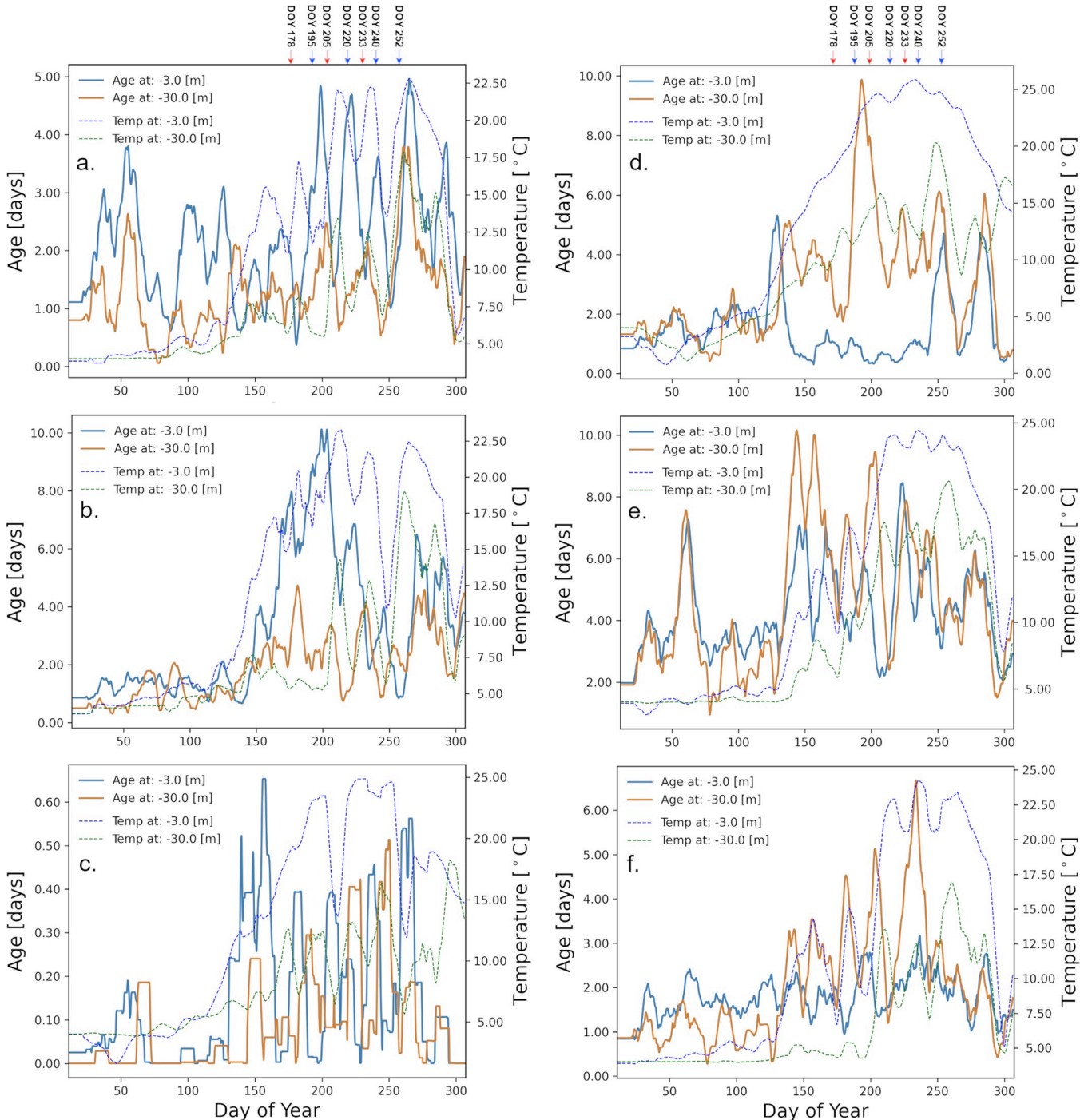

**Fig 7. Average age and temperature for nearshore sectors.** Daily average *age* (blue and dark orange lines, left axis) and temperature (blue dashed lines 2 m depth and green dashed line for 7 m, right axis). *Age* values are plotted with solid line, temperature values are plotted with dashed line. Note the scales of both y-axes are all different for better illustration purpose. The values were calculated as 10 days moving average for the entire 2018 simulation. (a) Humber Bay; (b) Hamilton-Burlington Basin; (c) Rochester Basin; (d) Wolfe Island—Kingston Basin (e) Presqu'ile Bay; (f) Oshawa (See Fig 1).

We looked at overall Pearson correlation coefficient *r*, between the wind and temperature time series with the *age* time series, as a measure of the influence of the wind speed over maximum annual *age* (Fig 8). The results indicate a good overall correlation (direct for temperature and

**Table 3. Shelf widths and bottom slopes for the nearshore sectors.**

| Nearshore Sector | Average shelf width [km] | Average Bottom Slope [°] |
|---|---|---|
| Humber Bay | 2.8 | 30 |
| Hamilton-Burlington Basin | 6.6 | 10 |
| Niagara | 5.1 | 13 |
| Rochester Basin | 2.8 | 45 |
| Oswego Basin | 3.2 | 43 |
| Wolfe Island—Kingston Basin | 26.6 | 15 |
| Presqu'ile Bay | 6.2 | 18 |
| Oshawa | 4.8 | 17 |

inverse for wind) for regions where the maximum annual age is around 20 days (e.g., 50 to 70% at Hamilton-Burlington Bay; Fig 8C, 8D), and a much weaker correlation for regions with low (6–7 days) maximum annual *age* (e.g., 12 to 32% at Humber Bay; Fig 8A, 8B). The calculated time-lag between temperature and age averaged at 6 days (Fig 8B, 8F). Additionally, a time-lag of 4 days was observed between wind and age (Fig 8D and 8H). These results indicate that beside wind and stratification, there are other factors that influence horizontal mixing, such as local geomorphology (width and slope of the nearshore shelf, ridges, groins, etc.). Further research is required to comprehend the impact of local geomorphological features, including large ones like the Tommy Thompson Park peninsula, on nearshore-offshore mixing. They contribute to shaping the direction of the coastal currents and forming of local eddies that directly influence mixing and sediment transport.

We performed a rough calculation of the nearshore-offshore exchange (Fig 9) using only the S-N velocities from the vertical transect following the 30 m depth contour on the northern shore vertical transect between the locations represented by two red dots (Fig 2), Depending on the depth of the thermocline, the cross-shore contribution of hypolimnetic waters to the net daily flux changed between 50% and 100% during the upwelling events (e.g., DOY 220 and 240), and it is practically zero during downwelling events (e.g., DOY 204). During the unstratified season, the thermocline does not exist and there is no representation of the flux (Fig 9B) and its contribution to the exchange (Fig 9C). These results indicate that episodic upwelling events can occasionally supply significant nutrients, such as phosphorus, to nearshore areas where favorable growth conditions, including shallow, warm, and clear waters, may exist. Consequently, there results may help with lake nutrient management by considering the hypolimnetic contribution in addition to the traditional local inputs (e.g., water treatment plants, tributaries and agricultural run offs).

Coastal upwelling events significantly influence seasonal nearshore-offshore exchanges. For instance, during upwelling events #4 and #10 (Fig 2), depth-averaged cross-shore velocities of approximately 0.014 ms$^{-1}$ and 0.019 ms$^{-1}$, respectively, would result in cross-shore excursions of approximately 1.1 km and 1.7 km per day along the north shore, assuming horizontal transport is primarily driven by advection. Accurately assessing historical trends in nearshore-offshore exchanges necessitates long-term records of water temperature and wind dynamics in coastal regions. In this study, we present an estimate of nearshore upwelling in Lake Ontario based on the data acquired during the field season, which was also used for the calibration of the model. The observed increasing trend in the frequency of upwelling-favorable winds along the north shore of Lake Ontario aligns with recent findings that indicate a rise in extreme eastward wind events facilitating inter-basin exchanges in Lake Erie [38]. Further investigation into similar wind-induced events is essential for predicting intra-basin and inter-basin physical and biological processes in lakes and coastal oceans.

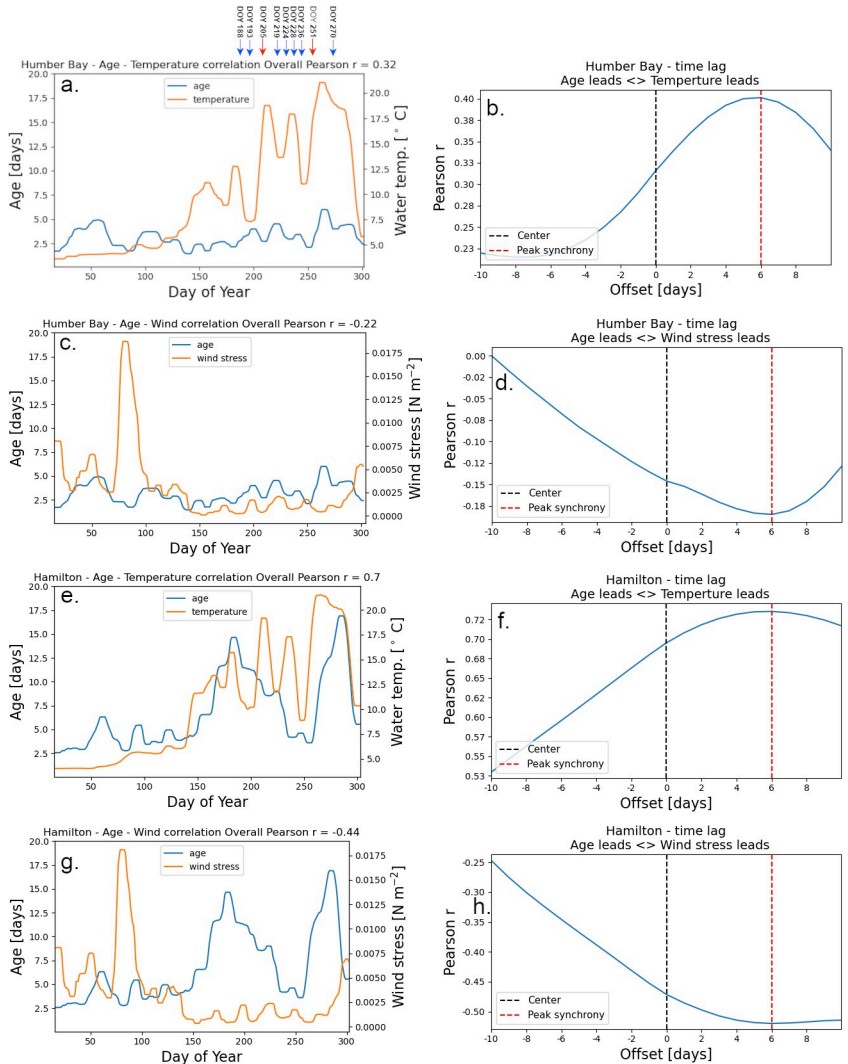

**Fig 8. Age- water temperature and age—wind speed correlation for 7 days moving average in the nearshore area.**
*Age* (blue) and *wind speed* and *temperature* (orange). (a) *age-temperature* correlation at the Humber Bay nearshore station as an example location with short age. (b) Determination of time lag between *temperature* and *age* using windowed time-lagged correlations. (c) *age-wind speed* correlation at the Humber Bay nearshore station. (d) Determination of time lag between *wind speed* and *age* at the Humber Bay nearshore station. (e) *age-temperature* correlation at Hamilton-Burlington Basin nearshore station as an example location with long age. The temperature of the epilimnion was used. (f) Determination of time lag between *temperature* and *age* using windowed time-lagged correlations at the Hamilton nearshore station. (g) *age-wind speed* correlation at the Hamilton nearshore station. (h) Determination of time lag between *wind speed* and *age* at the Hamilton nearshore station.

## Summary and conclusions

In this study, we employed a validated three-dimensional numerical model to replicate the circulation and thermal structure in the lake. We investigated horizontal mixing in Lake Ontario using the same three-dimensional hydrodynamic and water quality numerical model, augmented with virtual tracers. Two sets of modeling studies were conducted. The first involved simulating lake conditions over 2018, the same year used for model validation. This simulation examined water the horizontal mixing between the nearshore areas at the 30 m depth boundary and the offshore areas. The residence time of water in the nearshore areas was estimated

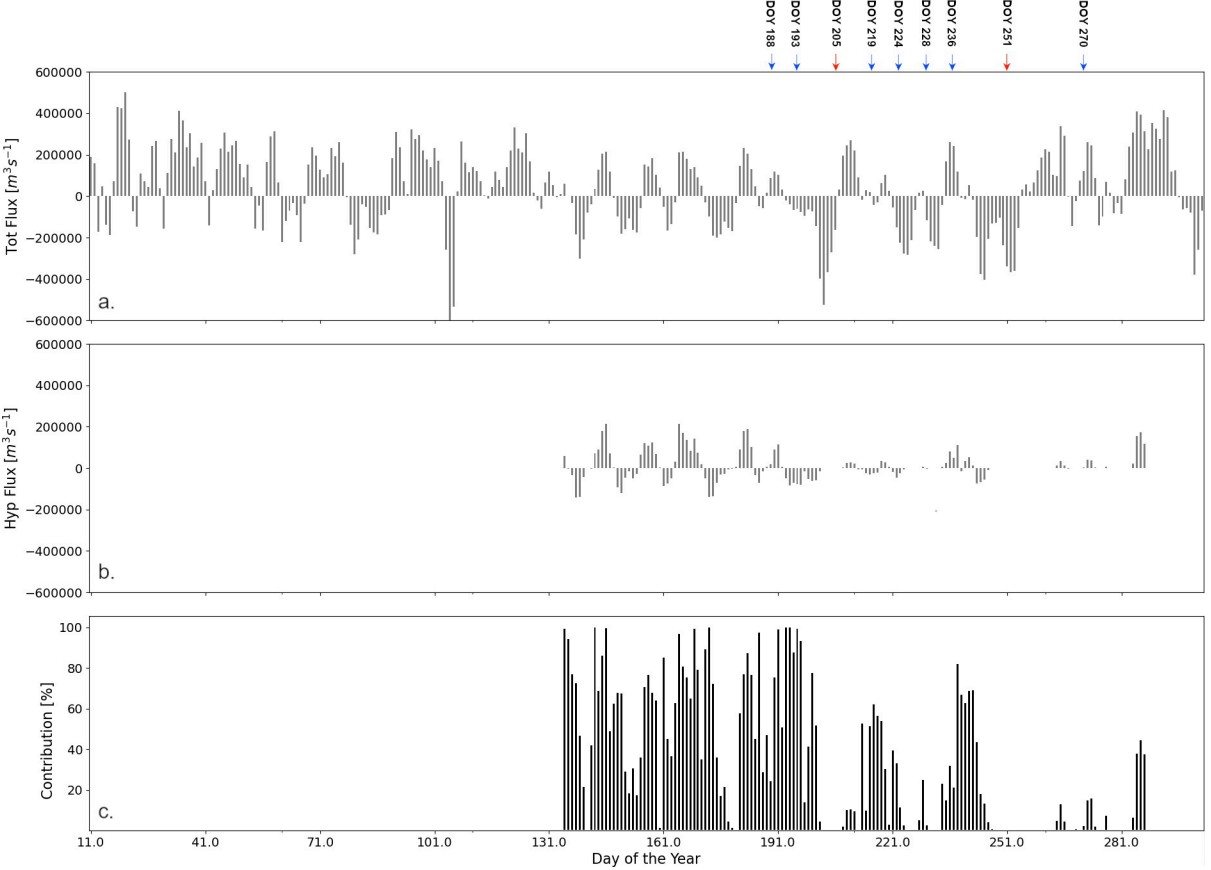

**Fig 9. Calculated total and hypolimnetic water south-north daily flux (+ to the north) through the vertical transect at 30 m depth contour, covering the northwestern basin between Burlington and Colborne (east of Cobourg, Ontario; see in Fig 2 the two red dots delimiting the transect).** (a) Total flux; (b) Hypolimnetic flux and (c) Hypolimnetic contribution (%) of the total exchange. At the top of the figure there are marked the most important upwelling (blue) and downwelling (red) events. The high contribution during the early summer can be explained by the shallow position of the thermocline that gives the majority of the water column depth for the hypolimnion.

using the *age* tracer. The horizontal mixing, inferred from changes in *age* estimates, was weaker during the isothermal period (fall and spring), when the water column was well-mixed [38].

To further understand the dispersal of nearshore water in the lake, we compared the dispersal of a passive tracer (*NRT*) released from 8 nearshore locations around the lake's perimeter in April, when the water column was isothermal and in July when the stratification was strong. We found that the dispersal was more intense in the stratified season, and more extensive cross-shore circulation was found at the western end of the lake basin. On the northern shore, the dispersal was primarily alongshore due to counterclockwise coastal circulation. Analysis of surface circulation showed that a large gyre formed in the central basin during low-wind periods following coastal upwelling events, whereas in the rest of the lake gyres were smaller in diameter. In the eastern basin, dispersal was limited to the southern shore most likely due to its geomorphology (steep and narrow shelfs). The analysis of horizontal mixing trough the proxy tracers showed that it was greater when average wind speed was higher, and when the nearshore shelf is narrower and steeper.

A second tracer study investigated exchange of the *NST* water with the offshore. A set of 4 simulations was completed with the open offshore open boundary condition set to a constant

100% *NST* concentration starting at different dates of the year, reflecting the diverse lake hydrodynamic and hydrostatic conditions. The results showed that the *NST* concentration in the offshore increased equally fast in both the upper and bottom layers during the isothermal periods. During the stratified period, the bottom layer was sheltered from wind action by the thermocline. Consequently, the *NST* concentration increased very slowly in the lower layers when compared to the surface layers. The increases in *NST* concentrations were consistent with the changes in *age* and indicated that the stratified season is more sensitive to water quality events, whether they are spills, tributary floods, or algae blooms.

Water exchange between the coastal areas and the offshore of large water bodies plays a crucial role in nutrient circulation. In the case of Lake Ontario, the connections mediated by water exchange are diverse although not well quantified. The challenges of nearshore eutrophication and offshore oligotrophication confronting the lake [27] hinge on water exchange [26, 27]. The proliferation of the benthic green algae *Cladophora* on the shallow lakebed around the lake [28, 39], concurrent with a decrease in open lake phosphorus concentrations to strongly oligotrophic levels not observed in decades [40], is believed to result from alterations in spatial patterns of nutrient transport influenced by dreissenid mussels, as proposed in the nearshore shunt hypothesis [9].

The flux of phytoplankton to the lower water column where large populations of mussels blanket much of the lakebed from the shoreline to deeper depths in the offshore [15], governs a pervasive component of the phosphorus movement within the lake. Dreissenid feeding on phytoplankton which disperse with the lake currents in the mixed surface layer, or the whole water column when the lake is mixed, moves biologically bound phosphorus to the lakebed, where it accumulates in living mussel biomass, and in waste products and decomposing biomass when mussels die. A widely speculated hypothesis is that the falling lakewide total phosphorus concentration is a result of increased storage of phosphorus on the lakebed since the dreissenid invasion from 1990 to 1995 and suggested in the most recent analysis of nutrient loads to the lake, which indicate lower water column concentration than predicted from external loads [40].

The patterns of movement of water within the lake likely influence the food supply to the mussel beds on a spatial organized manner. In the coastal areas, onshore movement of water and phytoplankton in the upper mixed layer of the open lake presents a substantial potential food supply by volume, although not as highly concentrated as in shoreline areas receiving nutrient discharges [40]. The tracer experiments, predicting water *age* in the nearshore indicate a substantial flux of water into the nearshore from the offshore during the stratified season across the 30 m depth contour (Fig 9), which is also suggested by the young age indicating dilution by offshore water. Notably, the strongest influence comes from the more forceful conditions during upwelling and downwelling events, with effects routinely evident in strong temperature swings along the coastline; downwelling events are likely more biologically relevant in terms of phytoplankton transport.

The generally greater flux into the nearshore at the lake surface compared with at 30 m depth, after lake stratification, indicates that onshore water movement need not benefit mussel food resources unless circulation into the coastal area is to a depth less than the mixed depth, which is highly variable, often in the range of 10 to 20m. Tracer experiments along the shoreline clearly indicate the prevalence of alongshore circulation consistent with horizontal mixing within the coastal boundary layer, extending to about 3 to 5 km offshore [19, 41–43]. The 30 m depth contour generally falls within 3 to 5 km offshore on the north shore and is generally closer to shore on the south side. Except for the Kingston Basin in the easternmost corner of the lake and to certain extent in the Niagara Basin in the southwest, the onshore exchange across the 30 m contour represents loading into the outer margin of the coastal boundary layer

for the most part. With knowledge of phosphorus load in the hypolimnetic waters and water exchange (Fig 9), one could estimate the nutrient flux transported from the offshore to the nearshore and evaluate its potential influence on the growth of nuisance algae.

In addition to algae, dreissenid mussels growing at shallow lake depth of <10-15m also have substantial access to phytoplankton transported from offshore waters. This has broad significance for phosphorus management in Lake Ontario, not only in terms of phosphorus loss offshore, but also as a possibly contributing factor in the proliferation of *Cladophora* and associated impacts, including fouling of beaches and water intakes. Offshore phosphorus transported onshore as phytoplankton opens up the nearshore phosphorus mass balance.

## Supporting information

**S1 File. Contains additional information about the model used, its calibration and validation and additional data.** The summary of the supplementary information file is listed below: Part I. Model Setup, Part II. Calibration and validation, Part III. Additional data, Part IV. Bottom slope analysis, Part V. Field Observation.
(DOCX)

## Acknowledgments

The authors would like to thank the vessel crew of the Great Lakes Monitoring Unit of the Ministry of the Environment, Conservation and Parks for the help with configuring, deploying, retrieving, and reading the field instruments used during this study.

## Author Contributions

**Conceptualization:** Bogdan Hlevca, Edward Todd Howell.

**Data curation:** Bogdan Hlevca.

**Formal analysis:** Mohammad Madani.

**Investigation:** Bogdan Hlevca, Edward Todd Howell.

**Methodology:** Reza Valipour, Mohammad Madani.

**Writing – original draft:** Bogdan Hlevca.

**Writing – review & editing:** Bogdan Hlevca, Edward Todd Howell, Reza Valipour, Mohammad Madani.

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
