## [Decision Letter · Decision Letter 0]

19 Mar 2024

PONE-D-24-03954Modeling nearshore-offshore water exchange in Lake OntarioPLOS ONE

Dear Dr. Hlevca,

Thank you for submitting your manuscript to PLOS ONE. After careful consideration, we feel that it has merit but does not fully meet PLOS ONE’s publication criteria as it currently stands. Therefore, we invite you to submit a revised version of the manuscript that fully addresses all the points raised during the review process by the two reviewers. Please submit your revised manuscript by May 03 2024 11:59PM. If you will need more time than this to complete your revisions, please reply to this message or contact the journal office at plosone@plos.org. Please include the following items when submitting your revised manuscript:A rebuttal letter that responds to each point raised by the academic editor and reviewer(s). You should upload this letter as a separate file labeled 'Response to Reviewers'.A marked-up copy of your manuscript that highlights changes made to the original version. You should upload this as a separate file labeled 'Revised Manuscript with Track Changes'.An unmarked version of your revised paper without tracked changes. You should upload this as a separate file labeled 'Manuscript'.

We look forward to receiving your revised manuscript.

Kind regards,

João Miguel Dias, Ph.D.

Academic Editor

PLOS ONE

Journal Requirements:

"This research was funded by the Ontario Ministry of the Environment, Conservation and Parks as part of its ongoing nearshore water quality monitoring and research."

5. In the online submission form, you indicated that "The datasets generated during the analysis phase are available from the corresponding author on reasonable request."

Reviewers' comments:

Reviewer's Responses to Questions

**Comments to the Author**

1. Is the manuscript technically sound, and do the data support the conclusions?

Reviewer #1: Partly

Reviewer #2: Yes

2. Has the statistical analysis been performed appropriately and rigorously? 

Reviewer #1: No

Reviewer #2: Yes

3. Have the authors made all data underlying the findings in their manuscript fully available?

Reviewer #1: Yes

Reviewer #2: Yes

4. Is the manuscript presented in an intelligible fashion and written in standard English?

Reviewer #1: Yes

Reviewer #2: Yes

5. Review Comments to the Author

Reviewer #1: Review of “Modeling nearshore-offshore water exchange in Lake Ontario”.

The manuscript describes the exchange dynamics between near- and offshore areas of Lake Ontario based on numerical simulations. The topic is interesting and the methodology followed is efficient. However, there are several points that require more clarifications and changes. I suggest major revisions based on my comments below.

General comment 1: The vertical structure of the water coloumn is not described at all. Only surface and deep temperature is shown. E.g., stratification frequency is presented only in supplementary material but it could be very useful to be included in the main manuscript.

General comment 2: The authors use the term “mixing” to express the horizontal advection or dispersion of tracers. Mixing usually refers vertical mixing. I suggest to correct this throughout the text.

General comment 4: The information about the different experiments/simulations is a bit confusing. The descriptions should be clarified (maybe with a use of a table?) and details about the runs’ differences (and the purpose for each simulation) should be included. E.g., I am confused about the release areas of tracers in the first and second simulation studies. Moreover, more information about the type and characteristics of the Lagrangian simulations is needed.

General comment 4: It is not clear, at least to me, why the authors chose to perform another set of simulation with the offshore boundary conditions and did not use the typical set up with releases at the 8 locations. This is maybe a problem of the simulations description (see above), which is a bit confusing.

Several specific comments (not in an order of importance) are provided below:

Specific comments

Chapter numbering is missing

L31: north or south?

Hievca et al. paper that presents the set-up and validation of the model used here is not yet published. It should be replaced with the published when/if it is published.

Fig. 1: A point at the westernmost tip of the lake does not have name.

L113-114: Please explain

Simulation of age: It is confusing to understand which of the two simulation studies is used here. I guess it is the second one but it should be clarified for all results. Please see also previous comment about the simulations’ description.

L152: I can’t see that. I see strong variation but not increasing trend after 135

L161-162: increase maybe? Shaded area cover a period of higher ages, no?

L168: I do not understand or even see this in the figure.

Fig. 2: Figure and caption need corrections. Labelled points? “The nearshore points of release are located at a water depth of less than 30 m.”. Are these presented in Fig. 2? Vertical transect?

Fig. 3:

- Please put legend in (b) and include all lines (temperature and ages) in both panels’ legends.

- Why not keeping the same color as age for temperature?

- Please align the two panels to facilitate viewing the changes in time between timeseries

- It is minor but why does the time x-axis stop a few days after 300 and not incudes the entire year? What is this “strange” dashed line right next to the left y-axis?

Table 1: Ages can also be included with a separate column and it should be described more in the text.

Simulation of nearshore regional tracers: So, are we still in the first simulation? It was stated that this simulation starts in April…How can we see results for entire year in Fig. 3? Please see previous comment about the simulations descriptions. Clarifications about the simulations/periods and which version is used for what reason/results is needed!

L198: Niagara is west or south?

L203: This is just an example. Please mark all topographic features, mentioned in the text, in figure 1.

L213: Same here

Fig. 4: L223-224: Which intervals? Averages for 62 days?

In all figures: use dates instead of only numbers. It is getting confusing (e.g., Fig. 5)

L250: This is the case in all scenarios. Please comment.

Fig. 5: Averaged over which areas exactly? Not sure what the shaded areas show.

L269: “a single year”. And even faster for the surface layer, no?

L271: “discussed above”. Where? In Fig. 3? Fig. 5 does not present ages.

Fig. 6: Please mark with a title on the panels the respective locations.

Table 2. Hamilton is missing in Fig. 1.

Fig. 7. r=-0.22 in 7c does not agree with the r in 7d. Please check.

Fig. 8. Discussion about this figure is nearly absent. More details are needed. What about the rest of the year that does not have any values in (b).

L343: “2 passive tracers”. Two sets? Two tracers?

L353: open…open

L381-383: This conclusion is not supported by the results. Please explain or remove.

L386-388. Same here.

Reviewer #2: In this manuscript, the authors used a three-dimensional hydrodynamic model along with a series of virtual tracer numerical experiments to investigate the water exchange between the nearshore and offshore areas of Lake Ontario. They found that cross-shore transport varies across different nearshore regions and is more pronounced during the stratification period than the water turnover period. Additionally, coastal upwelling events can contribute to cross-shore transport. These findings are meaningful as they enhance our understanding of nutrient transport and its connection to algae growth in the nearshore region.

However, the results and conclusions do not seem entirely convincing to me, as the manuscript lacks discussion on many aspects. It is also somewhat frustrating that all conclusions are based on simulations spanning only one year. Given these considerations, I recommend a major revision.

Major comments

1. One aspect that I think the study lacks discussion on is the impact of weather-scale wind events on cross-shelf transport in the top layer. Given the relatively shallow depth of the nearshore region, the magnitude and speed of weather-scale wind events can significantly influence its cross-shelf transport. For instance, Figure 4a shows strong alongshore dispersal in the northern nearshore, which the authors attribute to the cyclonic spin in the upper Mississauga Basin. However, it is possible that around DOY 62, before and after, a strong south wind occurred and inhibited water movement in the northern region. I think that an analysis of longer-term average currents and concentrations, along with a discussion on their variations in relation to weather-scale wind events, is necessary for a more comprehensive understanding.

2. I was wondering, did the hydrodynamic simulation include ice? If so, it is very interesting to see an enhancement of the cross-shore transport starting at DOY 30 in Figure 6a, coinciding with the time when ice begins to form more. Can you explain it a little bit? If not, please clarify it in the manuscript, as I suppose the nearshore-offshore water exchange will be very different in water with and without ice.

3. In the summary and conclusion, it is glad to see the author connecting the virtual tracer experiments with the nearshore shunt hypothesis regarding the nuisance conditions of Cladophora in the Lake Ontario nearshore area. The tracer experiments, suggested by the younger age indicators, point to dilution by offshore water, which transports particulate phosphorus (PP) to the nearshore area for mussel filtration. However, one aspect the author has overlooked is the simultaneous transport of soluble reactive phosphorus, excreted by mussels, to the offshore region through nearshore-offshore water exchange. This dual flux—water moving from offshore to nearshore and from nearshore to offshore—is typically not well balanced for individual cells or small regions. Consequently, in some areas, Cladophora may not grow to nuisance levels if the nearshore-offshore transport is significantly strong. Could the author elaborate more on this?

4. I think the term "mixing" as used in the manuscript is ambiguous and requires clearer definition. In most instances within the manuscript, it seems the authors are referring to the "mixing" as the integration of nearshore and offshore waters, primarily driven by horizontal advection. However, in several sections, "mixing" is used to denote diffusion processes.

Minor comments

1. I suggest adding one or two figures illustrating Lake Ontario's climatological circulation patterns and their seasonal variations in the introduction section to give readers a better understanding.

2. Lines 95-98: What about the water balance for the cell or grid? Can it cause an artificial plume in the cell and adjacent cells?

3. Line 224: There are no average currents shown in Figure 6.

4. Please add those stations or places (e.g., Oswego region, Humber Bay) to Fig. 4. Otherwise, it becomes difficult for readers to follow what you describe in the results section.

5. Lines 340-342: Lake Ontario is a dimictic lake. The water stratifies in winter and summer and is well-mixed in spring and autumn. Currently, the lake is not well-mixed during winter under the current climatic conditions….

6. PLOS authors have the option to publish the peer review history of their article (what does this mean?). If published, this will include your full peer review and any attached files.

Reviewer #1: No

Reviewer #2: **Yes: **Xing Zhou

---

## [Author Response · Author response to Decision Letter 0]

16 Sep 2024

The response to the reviewers has been attached as a separate file.

---

## [Decision Letter · Decision Letter 1]

22 Oct 2024

PONE-D-24-03954R1

Modeling nearshore-offshore water exchange in Lake Ontario

PLOS ONE

Dear Dr. Hlevca,

Thank you for submitting your manuscript to PLOS ONE. After careful consideration, we feel that it has merit but does not fully meet PLOS ONE’s publication criteria as it currently stands. Therefore, we invite you to submit a revised version of the manuscript that addresses the points raised during the review process.

We look forward to receiving your revised manuscript.

Kind regards,

João Miguel Dias, Ph.D.

Academic Editor

PLOS ONE

Reviewers' comments:

Reviewer's Responses to Questions

**Comments to the Author**

1. If the authors have adequately addressed your comments raised in a previous round of review and you feel that this manuscript is now acceptable for publication, you may indicate that here to bypass the “Comments to the Author” section, enter your conflict of interest statement in the “Confidential to Editor” section, and submit your "Accept" recommendation.

Reviewer #2: All comments have been addressed

2. Is the manuscript technically sound, and do the data support the conclusions?

Reviewer #2: Yes

3. Has the statistical analysis been performed appropriately and rigorously? 

Reviewer #2: Yes

4. Have the authors made all data underlying the findings in their manuscript fully available?

Reviewer #2: Yes

5. Is the manuscript presented in an intelligible fashion and written in standard English?

Reviewer #2: Yes

6. Review Comments to the Author

Reviewer #2: The authors have well addressed all my comments. Good job! I recommend this paper for publication. However, two minor things should be awarded before final publication:

1). The figures in the revised version are low resolution, making some labels difficult to read. Please ensure that high-resolution figures are included in the published version.

2). Fig. 2 and Fig. 3 should switch positions, as Fig. 3 is mentioned first in the Methods, while Fig. 2 is first mentioned in the Results."

7. PLOS authors have the option to publish the peer review history of their article (what does this mean?). If published, this will include your full peer review and any attached files.

Reviewer #2: No

---

## [Author Response · Author response to Decision Letter 1]

29 Oct 2024

The responses to the editor comments and reviewers have been submitted in an attached file.

---

## [Editor Report · Decision Letter 2]

31 Oct 2024

Modeling nearshore-offshore water exchange in Lake Ontario

PONE-D-24-03954R2

Dear Dr. Hlevca,

We’re pleased to inform you that your manuscript has been judged scientifically suitable for publication and will be formally accepted for publication once it meets all outstanding technical requirements.

Kind regards,

João Miguel Dias, Ph.D.

Academic Editor

PLOS ONE
---

## [Editor Report · Acceptance letter]

4 Nov 2024

PONE-D-24-03954R2 

PLOS ONE

Dear Dr. Hlevca, 

I'm pleased to inform you that your manuscript has been deemed suitable for publication in PLOS ONE. Congratulations! Your manuscript is now being handed over to our production team.

Kind regards, 

on behalf of

Prof. João Miguel Dias 

Academic Editor

PLOS ONE